# Highly Efficient Protoplast Isolation and Transient Expression System for Functional Characterization of Flowering Related Genes in *Cymbidium* Orchids

**DOI:** 10.3390/ijms21072264

**Published:** 2020-03-25

**Authors:** Rui Ren, Jie Gao, Chuqiao Lu, Yonglu Wei, Jianpeng Jin, Sek-Man Wong, Genfa Zhu, Fengxi Yang

**Affiliations:** 1Guangdong Key Laboratory of Ornamental Plant Germplasm Innovation and Utilization, Environmental Horticulture Research Institute, Guangdong Academy of Agricultural Sciences, Guangzhou 510640, China; renruinjau@163.com (R.R.); gaojie@gdaas.cn (J.G.); chuqiaolu18@163.com (C.L.); hjyylab@126.com (Y.W.); 13424050551@163.com (J.J.); 2Department of Biological Sciences, National University of Singapore (NUS), 14 Science Drive 4, Singapore 117543, Singapore; dbswsm@nus.edu.sg; 3National University of Singapore Suzhou Research Institute (NUSRI), Suzhou Industrial Park, Suzhou 215000, China; 4Temasek Life Sciences Laboratory, National University of Singapore, 1 Research Link, Singapore 117604, Singapore

**Keywords:** *Cymbidium* orchids, protoplast isolation, protoplast-based transient expression system (PTES), subcellular location, bimolecular fluorescence complementation (BiFC) assay, gene regulation

## Abstract

Protoplast systems have been proven powerful tools in modern plant biology. However, successful preparation of abundant viable protoplasts remains a challenge for *Cymbidium* orchids. Herein, we established an efficient protoplast isolation protocol from orchid petals through optimization of enzymatic conditions. It requires optimal D-mannitol concentration (0.5 M), enzyme concentration (1.2 % (*w*/*v*) cellulose and 0.6 % (*w*/*v*) macerozyme) and digestion time (6 h). With this protocol, the highest yield (3.50 × 10^7^/g fresh weight of orchid tissue) and viability (94.21%) of protoplasts were obtained from flower petals of *Cymbidium*. In addition, we achieved high transfection efficiency (80%) through the optimization of factors affecting polyethylene glycol (PEG)-mediated protoplast transfection including incubation time, final PEG4000 concentration and plasmid DNA amount. This highly efficient protoplast-based transient expression system (PTES) was further used for protein subcellular localization, bimolecular fluorescence complementation (BiFC) assay and gene regulation studies of flowering related genes in *Cymbidium* orchids. Taken together, our protoplast isolation and transfection protocol is highly efficient, stable and time-saving. It can be used for gene function and molecular analyses in orchids and other economically important monocot crops.

## 1. Introduction

Among the 350,000 plant species on earth, Orchidaceae (more than 25,000 species) is one of the largest and the most evolved families of monocot plants (Appendix A) [1,2]. Over the past 2000 years, more than 70,000 orchids have been cultivated as ornamental and medicinal plants, as well as a food flavoring additives [3]. Chinese *Cymbidium* orchids are regarded as an important symbol of oriental culture. To obtain high yielding quality orchid plants, intense efforts have been put into orchid biology research for the past few decades. Related researches include tissue culture, protoplast fusion and regeneration [4,5]. Recently, the release of orchid complete genome sequences of *Phalaenopsis* [6], *Dendrobium* [7] and *Apostasia* [8] has greatly facilitated gene cloning, genetic evolutionary analysis and synthetic biology. Advancements in plant genomics, transcriptomics, proteomics and metabolomics have renewed interest in orchid protoplasts. The versatility of protoplast-based platform helps biologists to explore intracellular processes involving cell structures [9], synthesis of pharmaceutical products [10] and toxicological assessments [11]. In recent studies, plant protoplasts have attracted great attention, since they can be isolated from specific plant tissues/organs and maintained their cellular identity, which practically provided a unique single cell system to answer specific questions related to cell types [12,13]. However, isolation of sufficient viable protoplasts remains a challenge for orchids and many other monocot crop plants [14].

Generally, young plant organs such as cotyledon, hypocotyl, leaf, root and root hair have been used as source materials for protoplast isolation [15]. Most common source of protoplast isolation is from leaf tissues. Plant mesophyll protoplasts have been successfully isolated from tobacco [16], *Arabidopsis* [17], maize (*Zea mays L.*) [18], rice (*Oryza sativa L*.) [19] and other non-model plant species, such as *Medicago sativa* [20], *Panicum virgatum* [21], *Elaeis guineensis* [22], *Hevea brasiliensis* [23], *Phaseolus vulgaris* [24] and Magnolia [25]. However, it is difficult to isolate protoplasts with high yield and viability from mature leaf tissues of many plants. Hence, young leaves of in vitro grown plantlets of *Phalaenopsis* [26], *Dendrobium* [27], *Tanacetum* [28], grape (*Vitis vinifera*) [29], pepper (*Capsicum annuum* L.) [30] and pineapple [31] have been used. Nevertheless, callus induction is time consuming and impractical for protoplast isolation. Hence, flower petals have been selected as an alternative for protoplast isolation for ornamental plants such as *Rosa rugose* [32], *Dendrobium* [33] and *Phalaenopsis* [34]. In addition to species types and source materials, enzyme combination and the concentration of osmotic pressure regulating substance D-mannitol also affect the effective release of protoplasts [22,35]. Therefore, an efficient method of protoplast isolation is still lacking in many non-model species. For family *Orchidaceae*, protoplasts are first successful isolated from leaves of *Cymbidium* [36] and various tissues of *Renantanda*, *Dendrobium* and *Paphiopedilum* [37,38]. Leaf tissues of in vitro grown seedlings of *Dendrobium* (3.97 × 10^5^/g fresh weight (FW)) [39] and *Cymbidium* (maximum 1.1 × 10^7^/g FW) [40], and flower petals of *Phalaenopsis* (1.1–5.9 × 10^6^/g FW) [34,41] have been used for successfully protoplast isolation. However, the protoplast isolation efficiency is lower than that of *Arabidopsis* (3.0 × 10^7^) [42], maize (1–5 × 10^6^/g FW) [43], rice (1.0 × 10^7^) [44] and cassava (4.4 × 10^7^/g FW) [45]. Up to now, few efficient protoplast isolation protocols have been established for orchids.

Protoplast-based transient expression system (PTES) has been proven to be a powerful experimental tool in molecular biology. It is widely used for protein localization, protein–protein interactions, identification of the gene function and gene regulation [14,34,44,46,47,48,49]. Transient protoplast transfection has been used due to its advantages in high screening efficiency with gene silencing and genome editing technologies, prior to the development of transgenic plants [43,50,51,52]. Protoplast transfection refers to the introduction of exogenous nucleic acids into protoplasts mediated by polyethylene glycol (PEG) or electroporation [53,54,55]. PEG-mediated transfection method is commonly used to introduce DNA or RNA into protoplasts. To obtain reliable and reproducible results, high transfection efficiency (>50%) is required [14]. In addition to viability and quantity of protoplasts, DNA or RNA amount, final concentration of PEG and incubation time are factors affecting transfection efficiency [41,56]. High level transfection efficiency (>80%) has been achieved using *Phalaenopsis* petal protoplasts [34]. This is equivalent to or better than that of previously reported transfection efficiencies [14,42]. However, the transfection efficiency dramatically decreases with an increase of plasmid DNA size from 5 to 10 kb [50,57,58,59]. Hence, improving transfection efficiency with a larger size plasmid is necessary for genome editing and other studies.

Limitations of existing protoplast isolation protocols led us to develop a simple and efficient orchid protoplast isolation and transfection method. This method recommends the use of flower petals and buds, which is readily available, sustainable and low cost. Using this protocol with optimal enzymatic conditions, high yielding viable protoplasts were successfully isolated efficiently from petals and buds of *Cymbidium orchids*. It does not require vacuum infiltration, treatment with high osmotic solution to plasmolyze tissues prior to enzymatic digestion, or special equipment. By optimizing factors affecting PEG-mediated protoplast transfection, we have also established a robust PTES for gene functional analysis. It takes less than 24 h from tissue preparation to confocal laser scanning microscopic observation or expression analysis using qRT-PCR. This PTES can be used for protein subcellular localization, protein–protein interaction and gene regulation analysis in *Cymbidium* orchids.

## 2. Results

### 2.1. Protoplast Isolation from Cymbidium Flower Petals

We aimed to establish an efficient protoplast isolation protocol for orchids. Herein, flower petals and buds from *Cymbidium* orchid plants were used (Figure 1A). To ensure that the released protoplasts do not rupture or collapse during enzyme digestion, different concentrations of D-mannitol (0.2, 0.3, 0.4, 0.5, 0.6 and 0.7 M) were tested. Viable protoplasts were released from petals with variable yield and viability (Figure 1B). Generally, the yields of protoplasts increased with increasing D-mannitol concentrations, and then decreased with increased osmotic pressures. The optimal D-mannitol concentration in enzyme solution for petals was determined as 0.5 M (Figure 1B). With optimal D-mannitol concentrations, orchid petals gave the highest yield (8.80 ± 0.68 × 10^6^/g FW) and viability (94.98% ± 1.51%) of petal protoplasts. In addition, protoplasts isolated from petals remained intact for more than 24 h in W5 solution. To increase the yield and viability of protoplasts, more factors affecting protoplast isolation including enzyme concentration and incubation duration were investigated. Given the combination of optimal D-mannitol concentration (0.5 M), enzyme mixture (1.2% cellulose and 0.6% macerozyme) and digestion time (6 h), the highest yield (3.50 × 10^7^ /g FW) and viability (94.21%) of *Cymbidium* protoplasts were achieved. The isolated protoplasts ranged from 20 to 100 μm in diameters, and a large proportion of protoplasts isolated from flowers petals were rich in cytoplasm and anthocyanidin. It indicates that our protoplast isolation protocol is suitable for the investigation of intracellular processes in tissues/organs-specific protoplasts of *Cymbidium* orchids.

### 2.2. PEG-Mediated Transient Expression in Orchid Protoplasts

In order to establish an efficient PTES for protein subcellular location, protein–protein interaction and gene regulation studies (Figure 2A), factors affecting PEG-mediated protoplast transfection were investigated. The transfection efficiency increased from 45.26–75.39% within 5–15 min, but decreased from 75.69% to 51.11% within 20–25 min (Figure 2B), indicating that the optimal incubation time was 15 min. The transfection efficiency increased from 29.19% to 79.09% with increasing PEG4000 concentration, and then decreased from 79.09% to 62.42% when the incubation time was 15 min, indicating optimal final PEG4000 concentration was 25%. Moreover, when given the optimal incubation time (15 min) and the final PEG4000 concentration (25%), the transfection efficiency increased significantly when using 1–2 μg/μL of plasmid DNA, and the efficiency remained at high level (80%). To sum up, factors affecting transfection efficiency were optimized as follows: transfection duration (15 min), PEG4000 concentration (25%) and 1 μg/μL of plasmid DNA. A high transfection efficiency of 75–80% was achieved in orchid protoplasts.

A small amount of plasmid DNA was sufficient to allow fluorescence observation of fusion proteins in protoplasts (Figure 2C). A low concentration of 1 μg/μL of plasmids was used for subsequent protoplasts transfection. Using the optimized transfection method, a larger size plasmid (12 kb) pCambia1301-GFP was transformed, and the fluorescence of green fluorescent protein (GFP) expressed by the vector was detected in 30% of the transfected protoplasts (Figure 2C).

### 2.3. Protein Subcellular Localization

To test the feasibility of the protoplast transient expression system, we investigated the subcellular localization of the two flowering related homologous genes *APETALA3* (*CsAP3*) and *PISTILLATA* (*CsPI*) in *Cymbidium* orchid using our PTES. Recombinant vectors used for subcellular localization of fusion proteins were obtained by cloning the full-length coding sequences (CDSs) of *CsAP3* and *CsPI* into the transient expression vector PAN580-GFP (Figure 3A). Recombinant vectors pAN580-CsAP3-GFP and pAN580-CsPI-GFP, as well as empty control vector (expressing GFP) were transformed into *Cymbidium* protoplasts. Twelve to sixteen hours post transfection (hpt), green fluorescence was observed in intracellular compartments of transfected protoplasts. The vector control was distributed throughout the entire cell, while fusion proteins CsAP3-GFP and CsPI-GFP were both colocalized with the 4’-6’-diamidino-2-phenylindole (DAPI) signal, indicating nuclear localization of both fusion proteins (Figure 3B).

### 2.4. Protein–Protein Interaction Studies in Cymbidium Protoplasts

Yeast cells cotransformed with vectors GAD-CeAP3 and GBK-CePI gave blue colonies growth on QDO/X/A medium consistent with that of positive control rather than negative control (Figure 4A), which indicated that CsAP3 and CsPI proteins interact with each other (Figure 4A). Using this PTES protocol, protein–protein interaction was confirmed by bimolecular fluorescence complementation (BiFC) assay. Vectors used for BiFC assays were obtained by cloning the full-length CDSs of *CsAP3* and *CsPI* into vectors pSPYNE-35S and pSPYCE-35S, respectively (Figure 4B). Fusion proteins CsAP3-YFPn and CsPI-YFPc were co-expressed in *Cymbidium* protoplasts and yellow fluorescent protein (YFP) signals were detected in the nuclei where the DAPI signal was also present. As negative controls, combinations of empty YFPn + CsPI-YFPc, CsAP3-YFPn + empty YFPc and empty YFPn + YFPc did not produce any BiFC fluorescence (Figure 4C). In addition, BiFC assays were also carried out using another transient expression method in *N. benthamiana* leaves. As expected, strong YFP signals were observed in *N. benthamiana* mesophyll cells, especially nuclei (Figure 4D), which confirmed the interaction between proteins encoded by *CsAP3* and *CsPI*. Taken together, the PTES protocol is suitable for protein subcellular localization and protein–protein interaction studies.

### 2.5. Gene Regulation Analysis in Cymbidium Protoplasts Using qRT-PCR

To identify functions of flowering pathway integrators, expression of *Cymbidium* homologous of *FLOWERING LOCUS T* (*CsFT*) and *SHORT VEGETATIVE PHASE* (*CsSVP*) to their downstream genes were investigated in *Cymbidium* protoplasts by qRT-PCR. Using the PTES, *CsFT* was successfully transient overexpressed in *Cymbidium* protoplasts, which in turn enhanced the expression of *SUPPRESSOR OF OVEREXPRESSION OF CONSTANS1* (*CsSOC1*), *CsAP1*, *LEAFY* (*CsLFY*) and *CsSVP* expression (Figure 5A). Among the four genes, *CsSOC1* was the only significantly up-regulated gene with 5–16 folds at 12–36 hours post transfection (hpt). Similarly, *CsSVP* and *CsAP1* were both significantly upregulated for 2–5 folds from 24–36 hpt. The expression of *CsLFY* was significantly upregulated only at 12 hpt, but was significantly downregulated from 12–36 hpt. These results suggested that the transient overexpressed *CsFT* enhanced the expression of *CsSOC1*, *CsAP1* and *CsSVP*. In addition, *CsSVP* was also successfully overexpressed in *Cymbidium* protoplasts 12–36 hpt, and the expression levels of *CsSOC1*, *CsAP1*, *CsLFY* and *CsFT* were all significantly suppressed (Figure 5B). The expression of *CsSOC1* and *CsFT* were both significantly downregulated 12–36 hpt. The expression levels of *CsAP1* were downregulated at 12–36 hpt, while that of *CsLFY* were downregulated from 24–36 hpt, respectively. These results indicated that *CsSVP* suppressed the expression of *CsSOC1*, *CsAP1*, *CsLFY* and *CsFT*. The above results indicated that our PTES could be used to explore the inter-regulation relationship among orchid genes.

## 3. Discussion

Previous attempts to isolate higher yield and viability of protoplasts from different tissues of numerous species have been made separately under different conditions [14,34,42]. Though vacuum infiltration or treatment with high osmotic solution of plasmolyzed tissue prior to the enzymatic digestion has been recommended [14], high yielding viable protoplasts were obtained from flower petals of *Cymbidium* orchids without those treatments. In this study, we developed a simple, efficient and low cost orchid protoplast isolation method through optimizing enzymatic conditions (Figure 1). Osmotic conditions significantly influenced the yield of viable protoplasts (Figure 1B), which is in agreement with previous reports [14,16,18,46,60]. Plant cells are surrounded by cell walls and cytoplasm maintains an osmotic balance through vacuoles. Protoplasts are plant cell whose cell walls are enzymatically removed using cellulose-R10 combined with macerozyme-R10 [14,34]. We noted that increase in total concentration of enzymes led to an increase in protoplast yield, while excess enzymes led to a decrease in protoplast yield and viability (data not shown), probably due to phytotoxicity of enzymes on the membrane of protoplasts [61,62]. To obtain a higher number of viable protoplasts, a suitable concentration of additional enzymes such as pectolyase Y-23 [63] or/and hemicellulose may be added [24,29]. The efficiency of protoplast isolation from orchids was greatly increased from 10^6^ to 10^7^/g FW (almost ten folds) with this protocol. Given the optimal digestion conditions, sufficient protoplasts were obtained from flower petals and buds of *Cymbidium*, which is higher efficient than that of past.

In addition to protoplast fusion, the PTES is also an important experimental tool in plant molecular biology (Figure 2A). Protoplast transfection efficiency >50% is required for reliable and reproducible experimental data [14]. By optimizing factors affecting PEG-mediated transfection (Figure 2B), we achieved high protoplasts transfection efficiency (80%), which is equivalent or greater than that reported in other systems [34,41]. Previously PEG and calcium ion (Ca^2+^) are added to increase the fluidity of plasma membrane [53] to facilitate plasmid DNA uptake in protoplasts. Although the transfection efficiency increased with higher PEG4000 concentration and incubation period, the number of ruptured protoplasts increased after the threshold was reached (Figure 2B). Hence, suitable concentration of PEG and incubation time should be fully considered when establishing an efficient protoplasts transfection protocol. Higher transfection efficiency was obtained with a higher quantity of plasmid DNA (Figure 2B), which is consistent with other protocols [41,49]. In consideration of simple small-scale plasmid DNA isolation for fluorescence observation of fusion proteins in protoplasts (Figure 2C), a lower concentration of 1 μg/μL was used for subsequent protoplasts transfection. In addition, a larger size plasmid (12 kb) was transformed using the optimized transfection method, and the fluorescence of GFP reporter expressed by the vector was detected in 30% of protoplasts (Figure 2C). It provides wide application of our protoplast transfection system for rapid screening of targets of genome editing and gene silencing.

The efficient PTES provides a great platform for gene function identification. It relies on transient expression of transgenes rather than stable transgene in a given plant species. Hence, high throughput PTES are used for rapid screening of transactivation of hundreds of transcription factors [64] and rapid screening of numerous transgenic expression cassettes prior to stable plant transformation [49]. Here, the PTES was demonstrated for protein subcellular localization, BiFC assay and gene regulation analyses of flowering related genes (Figure 3, Figure 4 and Figure 5). As protoplast isolated from certain plant organs or tissues maintain their cellular identity, protoplasts isolated from petals and buds exhibit greatly feasibilities in functional characterization of flowering related genes in orchids and other ornamental plants. The B-class genes *AP3* and *PI* are of particular importance in the specification of petal and stamen identity [65,66,67]. In the present study, both *CsAP3* and *CsPI* were co-localized in nuclei of *Cymbidium* protoplasts (Figure 3). The result was consistent with that of Y2H assay (Figure 4A), and was confirmed with the BiFC assay in *N. benthamiana* leaves. In addition, we investigated the regulation of the downstream genes of *CsFT* and *CsSVP* in *Cymbidium* protoplasts (Figure 5) and the results were consistent with previous published results [68,69,70]. It indicated that genetic pathways mediated by FT and SVP (Figure 6) were conserved in orchids and other plant species [70]. These results reinforced the high reliability of the PTES in cellular and molecular biology research for orchids.

## 4. Materials and Methods

### 4.1. Plant Materials

*Cymbidium* orchid plants were obtained from the orchid breeding base of Environmental Horticulture Research Institute, Guangdong Academy of Agricultural Sciences, China. They were maintained in plastic pots (20 × 20 cm) in greenhouses with environmental conditions as previously described [71]. In addition, tobacco (*N. benthamiana*) plants were also included. Tobacco seeds were obtained from the National Key Laboratory for Crop Genetics and Germplasm Enhancement, China. The tobacco seedlings were grown in plant growth rooms at 27 °C under a 16 h light/8 h dark cycle with 60% humidity.

### 4.2. Protoplast Isolation

The protoplast isolation was developed from modifying protocols previously established [14,34,40,41]. Flower petals and buds from *Cymbidium* were collected for protoplast isolation. The tissues were cut into 0.51 mm strips using fresh surgical blades on sterile filter papers, and were immediately transferred into the freshly prepared enzyme-solution in a 100 mL sterile flask. Generally, 10 mL enzyme solution would be prepared for approximately 0.5 g tissues FW for each treatment. The enzyme solution was prepared as follows: 20 mM KCl (Sigma, St. Louis, MO, USA), 20 mM 2-(N-Morpholino)ethanesulfonic acid (MES, pH = 5.7; Sigma, St. Louis, MO, USA) with different concentrations of D-mannitol (0.2, 0.3, 0.4, 0.5, 0.6 and 0.7 M; Sigma, St. Louis, MO, USA), cellulose R-10 (1.2%, *w*/*v*; Yakult Pharmaceutical Industry Ltd., Nishinomiya, Japan) and macerozyme R-10 (Yakult Pharmaceutical Industry Ltd., Nishinomiya, Japan; 0.6%, *w*/*v*). The solution was then warmed up to 55 °C for 10 min, and cool to room temperature (25 °C). After that, 10 mM CaCl_2_ (Sigma, St. Louis, MO, USA) and 0.1% (*w*/*v*) bovine serum albumin (BSA; Sigma St. Louis, MO, USA) were added.

The released protoplasts were harvested after incubation at 28 °C in dark with a rotation of 30 rpm for different period of time (2, 4, 6 and 8 h). The enzyme mixture was diluted with equal volume of wash solution (W5) that containing 154 mM NaCl (Sigma, St. Louis, MO, USA), 125 mM CaCl_2_, 5 mM KCl and 4 mM MES (pH = 5.7). The protoplast-containing solution was filtered through a 150 μm nylon mesh into a 50 mL round-bottomed centrifuge tube. The flow through was centrifuged at 100× *g* for 2 min (at room temperature) to pellet the protoplasts, and the supernatant was removed carefully with a sterile syringe. The protoplasts pellet was resuspended gently in 10 mL W5 solution [72] and re-centrifuged at 100× *g* for 2 min (at room temperature), followed by resuspending in 5 mL W5 solution and incubated on ice for 30 min, the protoplasts were settled at the bottom of the tube by gravity. Finally, the supernatant was carefully decanted and discarded, and the purified protoplasts were adjusted to a density of 1 × 10^5^–1 × 10^6^/mL with prechilled MMG solution [73] containing 0.4 M mannitol, 15 mM MgCl_2_ and 4 mM MES at pH 5.7.

The protoplasts were counted and photographed by a Leica DM2500 microscope (Leica, Wetzlar, Germany) with a hemocytometer for calculation of estimated yield of protoplasts. The protoplasts yield was measured as the total number of protoplasts released in the enzyme mixture divided by the fresh weight of the tissues used for protoplast isolation (protoplasts/g FW). Fluorescein diacetate (FDA; Sigma, St. Louis, MO, USA) stain was used to determine the viability of protoplasts as follows: 9 μL of the protoplasts were put onto a glass slide followed by 1 μL of 0.2% FDA solution (dissolved in acetone (Solarbio Science and Technology Co., Ltd. Beijing, China)), and incubated at room temperature for 1–2 min. The viable protoplasts with green fluorescence were visualized and photographed by LSM 710 confocal laser microscope (Carl Zeiss, Inc., Jena, Germany) with blue excitation block. The protoplasts viability was measured as green cells/ total cells × 100%. Three images were selected for yield and viability measurement of each sample. Each experiment was repeated three times.

### 4.3. PEG-Mediated Protoplast Transfection

Protoplast transfection was carried out following a modified PEG-mediated protocol [14] (Figure 2A). The protoplasts were adjusted to 1 × 10^5^–1 × 10^6^ /mL in density with MMG solution. For each transformation, 20 μL prechilled plasmid DNA with different concentrations (0.25, 0.50, 1.00, 2.00, 3.00 and 4.00 μg/μL) was gently mixed with 200 μL protoplasts in 2 mL round-bottomed centrifuge tubes. Then equal volume (220 μL) freshly prepared PEG-CaCl_2_ solution was immediately added and mixed completely by gently inversion. The PEG-CaCl_2_ solution was comprised of 100 mM CaCl_2_ and 0.2 M D-mannitol with different final concentrations of PEG4000 (Sigma, St. Louis, MO, USA; 10%, 20%, 30% and 40%, *w*/*v*). The mixture was incubated at room temperature in the darkness with different durations (5, 10, 15, 20 and 25 min). The transfection was stopped by adding two volumes (880 μL) of W5 solution followed by centrifugation at 100 × *g* for 2 min. The transfected protoplasts were then washed with W5 solution and re-suspended with 1 mL WI solution in each of the 6-well tissue culture plate. The WI solution was comprised of 0.4 M D-mannitol, 20 mM KCl and 4 mM MES (pH 5.7). For transient expression of proteins (genes), the protoplasts mixture was incubated for 12–36 h at 23 °C in the darkness.

Transformation efficiency of the protoplasts was detected according to the expression of GFP reporter using the transient expression vector pAN580-GFP and the binary expression vector pCambia1301-GFP. The GFP fluorescence was observed and 3–5 images were taken in random distribution under LSM780 fluorescent microscope (Carl Zeiss, Inc., Jena, Germany) or LSM710 confocal laser scanning microscope. The transformation efficiency was measured as bright green fluorescent protoplast number in view/total protoplast number in view (%). At least three photographs were taken for each sample, and these experiments were independently conducted at least three times.

### 4.4. Protein Subcellular Localization

Cellular characters of proteins encoded by *CsAP3* and *CsPI* were investigated using protein localization with this orchid PTES. Vectors used were obtained by cloning the coding sequences (CDSs) of CsAP3 (GenBank accession number: JQ326260.1) and *CsPI* (JQ326259.1) into the vector PAN580-GFP. Briefly, total RNA was extracted from freshly collected young leaves of *Cymbidium sinense* using an RNA Simple Total RNA Kit (Tiangen, Beijing, China). The DNA-free RNA was used for first-strand cDNA synthesis with oligo (dT) primers and a PrimeScript™ 1st strand cDNA Synthesis Kit (Takara, Dalian, China), following the manufacturer’s instructions. The specific primers with overlapping homologous ends were designed using Primer Premier 5.0 software (Premier, Palo Alto, CA, USA) according to the full-length CDSs of *CsAP3* and *CsPI*, respectively (Appendix A). Subsequently, the fragments for each gene were amplified using PrimerSTAR Max DNA Polymerase (Takara, Dalian, China) with the cDNA and the specific primers. The PCR products were purified and full-length CDSs without termination codon were cloned into certain vectors by recombination using the Seamless Assembly Cloning Kit (CloneSmarter, Houston, USA) following the manufacturer’s instructions. Recombined vectors were transformed into Escherichia coli DH5α competent cells (Tiangen, Beijing, China) according to the manufacturer’s instructions and confirmed by sequencing. Ultimately, CDSs of *CsAP3* and *CsPI* were inserted between the dual cauliflower mosaic virus (CaMV) 35S promoter and GFP gene of pAN580-GFP for subcellular localization analysis, resulting in the recombination plasmids of pAN580-CsAP3-GFP and pAN580-CsPI-GFP expressing GFP fusion-proteins, respectively.

The above mentioned vectors including pAN580-GFP, pAN580-CsAP3-GFP and pAN580-CsPI-GFP were transformed into *Escherichia coli* DH5α competent cells (Tiangen, Beijing, China) according to the manufacturer’s instructions. Following mass replication of the bacterium, plasmid DNA was extracted by Endo-Free Plasmid Maxi Kit (Omega Bio-tek, Norcross, USA). The concentrated plasmid DNA was prepared with different concentrations (up to 2.0 μg/μL), and was transformed into the orchid protoplasts. After incubation at 23 °C in the darkness for 12–16 h, the fluorescence of GFP or GFP-proteins fusions were viewed under LSM710 confocal laser scanning microscope. To detect cell nucleus, the transfected protoplasts were stained with 50 μg/mLDAPI (Sigma–Aldrich Chemie, Steinheim, Germany) at 37 °C for 10 min. DAPI signals were excited with a blue diode laser (405 nm excitation line with 485-nm long-pass barrier filter).

### 4.5. Yeast Two-Hybrid Assay

The interaction between proteins encoded by *CsAP3* and *CsPI* was investigated by a yeast two-hybrid assay. In the same way, the full-length CDSs of *CsAP3* and *CsPI* were cloned to vectors pGADT7 and pGBKT7, resulting into recombination plasmids GAD-CsAP3 and GBK-CsPI, respectively. Fusion plasmids GAD-CsAP3 and GBK-CsPI were transformed into Y187 and Y2HGold yeast strains, respectively. Diploids containing GAD-CsAP3 and GBK-CsPI were obtained according to the small-scale protocol (Clontech, Palo Alto, USA). After selection of cotransformants on DDO (SD/-Trp/-Leu) medium, they were transferred to QDO/X/A (SD/-Trp/-His/-Trp/-Ade/X-α-gal/AbA) medium. Yeast cells carrying the pGBKT7-53 and pGADT7-T plasmids were served as the positive control, while yeast cells harboring the pGBKT7-Lam and pGADT7-T plasmids were used as the negative controls. Blue colonies were observed on QDO/X/A plates within 3–5 days, once the two proteins (encoded by *CsAP3* and *CsPI*) were interacted with each other.

### 4.6. BiFC Assay in Orchid Protoplasts and N. benthamiana Leaves

The vectors used for BiFC assay were prepared accordingly. Empty vectors pSPYNE-35S (harboring the nitrogen terminal of the yellow fluorescent protein (YFP)) and pSPYCE-35S (harboring the carbon terminal of the YFP) were both driven by the CaMV 35S promoter. Full-length CDSs of *CsAP3* and *CsPI* were cloned into pSPYNE-35S and pSPYCE-35S, resulting in recombinant plasmids pSPYNE-35S-CsAP3-YFPn and pSPYNE-35S-CsPI-YFPc, respectively (Figure 4B). For BiFC essay, pSPYNE-35S-CsAP3-YFPn and pSPYNE-35S-CsPI-YFPc were cotransfected into protoplasts. As negative controls, vector combinations pSPYNE-35S-CsPI-YFPc + pSPYNE-35S, pSPYCE-35S + pSPYNE-35S-CsAP3-YFPn and pSPYNE-35S + pSPYCE-35S were also cotransfected into the protoplasts, respectively. BiFC assays were carried out in *N. benthamiana* leaves to confirm the interaction of proteins encoded by *CsAP3* and *CsPI*. Briefly, the vectors pSPYNE-35S-CsAP3-YFPn and pSPYNE-35S-CsPI-YFPc were transfected into *Agrobacterium* tumefaciens strain GV301, respectively. Positive agrobacteria which fused with reciprocal halves of YFP were co-infiltrated into *N. benthamiana* leaves, and the *N. benthamiana* were cultured in the darkness for two days. The transfected protoplasts and the infiltrated leaves were examined under LSM710 confocal laser scanning microscope. The interaction was confirmed using both combinations of reciprocal YFPn/YFPc fusion proteins in two separate experiments (with three replicates per experiment).

### 4.7. Gene Regulation Analysis by qRT-PCR

The gene *FT* is the "flowering element" of plants that induces them to bloom through forming flowering complexes with a series of proteins with transcriptional activity [68,69,70]. However, *SVP* genes suppress the expression of flowering integration factors *SOC1* (MF474250.1), *FT* and *LFY* (KC138806.1) [68,69,70]. Therefore, these two key flowering regulatory genes *CsFT* (accession number: HM803115.1) and *CsSVP* (MF462098.1) were chosen to investigate the regulations to their downstream genes in orchid protoplasts. Full–length CDSs of *CsFT* and *CsSVP* were cloned into vector pAN580, and replaced with GFP gene, resulting in plasmids pAN580-CsAP3 and pAN580-CsPI, respectively (Appendix A). Both plasmids transiently expressing proteins encoded by *CsFT* and *CsSVP* were transfected into *Cymbidium* protoplasts, respectively. The protoplasts were incubated at 23 °C in the dark and harvested at 12, 24 and 36 hpt. The relative expression levels of *CsFT* and *CsSVP* along with their downstream genes *CsLFY*, *CsSOC1* and *CsAP1* were determined by qRT-PCR.

### 4.8. qRT-PCR

The transfected protoplasts (5 × 10^5^ protoplasts) were independently collected with three biological replicates and stored at -80 °C immediately after freezing into liquid nitrogen. Total RNA was extracted and treated with RNase-free DNase according to the manufacturer’s instruction. RNA was reverse-transcribed using PrimeScript™ RT reagent Kit with gDNA Eraser (Takara, Dalian, China) following the manufacturer’s instructions. The resulting first-strand cDNA was subjected to relative qPCR as previously described [71]. Gene-specific primers were designed from transcript sequences of *CsFT*, *CsSVP*, *CsLFY*, *CsSOC1* and *CsAP1* [74], using Primer-BLAST software (https://www.ncbi.nlm.nih.gov/tools/primer-blast/; Appendix A). The gene ubiquitin in *Cymbidium* (referred to as *CsUBQ*, accession No: AY907703) was used as an internal reference control to normalize the total amount of cDNA in each reaction (Appendix A). PCR was conducted, and only primers that amplified a single product were selected for qRT-PCR. The qRT-PCR was performed in a 20 μL reaction volume comprising 2.0 μL (20 ng) of 5× diluted first-strand cDNA, 0.8 μL of each primers (10.0 μM), 10.0 μL of 2× SYBR Green I Master Mix (Takara, Dalian, China) and 6.4 μL of sterile distilled H_2_O. All reactions were performed in 96-well reaction plates using a Bio-Rad CFX-96 Real-time PCR System (Bio-Rad, Hercules, Canada) with three technical replicates. The following PCR conditions were used: 95 °C for 5 min, followed by 40 cycles at 95 °C for 15 s, 60 °C for 30 s and 72 °C for 30 s, and then at 68 °C for 5 min. The expression of candidate genes was quantified using the relative quantification (2^−ΔΔCT^) method [75].

### 4.9. Statistical Analysis

The statistical analysis was performed with SPSS Version 18.0 software (SPSS Inc., Chicago, IL, USA). All experiments were replicated three times. Data are presented as mean–standard error from three independent experiments. Significant differences among treatments were determined at *p* ≤ 0.05 based on the Tukey–Kramer test.

## 5. Conclusions

The protoplast isolation and transfection system described in this study was proven to be robust, reliable and sustainable. As a time-saving tool-kit, it has a wide application, which includes virus replication, protein localization, protein–protein interactions, gene regulation and gene function identification in orchids.

## Figures and Tables

**Figure 1 ijms-21-02264-f001:**
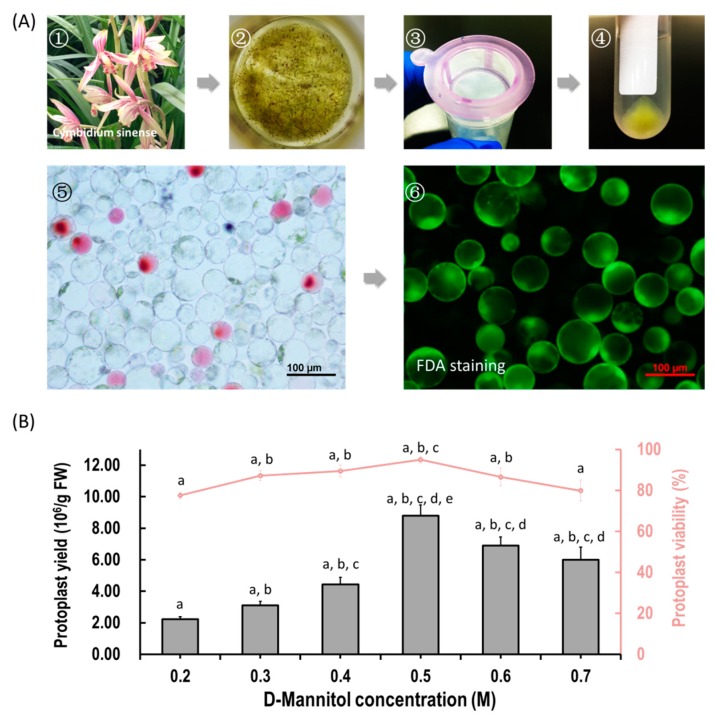
Protoplast isolation from flower petals of *Cymbidium* orchids. (**A**) Schematic illustration of protoplast isolation from *Cymbidium* orchids. (1) *Cymbidium* flower petals were collected and cut into 0.5–1.0 mm strips using fresh surgical blades on sterile filter papers; (2) immediately, transfer strips into in a 100 mL flask containing enzymatic solution; (3) after 5 h incubation at 28 °C in dark, the solution containing released protoplasts were filter through a 150 μm nylon mesh into a 50 mL round-bottomed tube; (4) following centrifuge at 200 rpm for 2 min, the protoplast pellet were re-suspended with W5 solution; (5) check the morphology and yield of protoplasts under a microscope and (6) measure the protoplast viability using the FDA staining method. (**B**) Optimization of *Cymbidium* protoplast isolation conditions. A series of D-mannitol concentration in the enzyme solutions (0.2, 0.3, 0.4, 0.5, 0.6 and 0.7 M) were tested for different tissues to ensure that released protoplasts do not rupture or collapse during enzyme digestion; data presented as means of three biological replicates with error bars indicating standard deviations (SD), and different letters (a–e) among treatments indicate statistically significant differences at *p* < 0.05 based on the Tukey–Kramer test.

**Figure 2 ijms-21-02264-f002:**
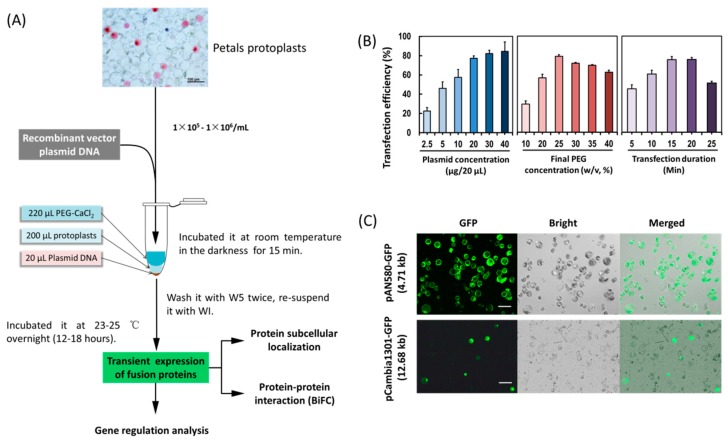
Establishment of an efficient protoplast-based transient expression system (PTES) for *Cymbidium* orchids. (**A**) Schematic illustration of protoplast transfection and application of the PTES; (**B**) factors affecting the PEG-mediated protoplast transfection, including incubation time, final PEG4000 concentrations and plasmid DNA amount were optimized and (**C**) the green fluorescence of the green fluorescent protein (GFP) reporter expressed by the two vectors was clearly visible. The maximum transfection efficiency of PAN-580 (4.71 kb in size) and pCambia1301-GFP (12.68 kb) were 80% and 30%, respectively. Data presented as means of three biological replicates with error bars indicating the SD. Bar = 100 μm.

**Figure 3 ijms-21-02264-f003:**
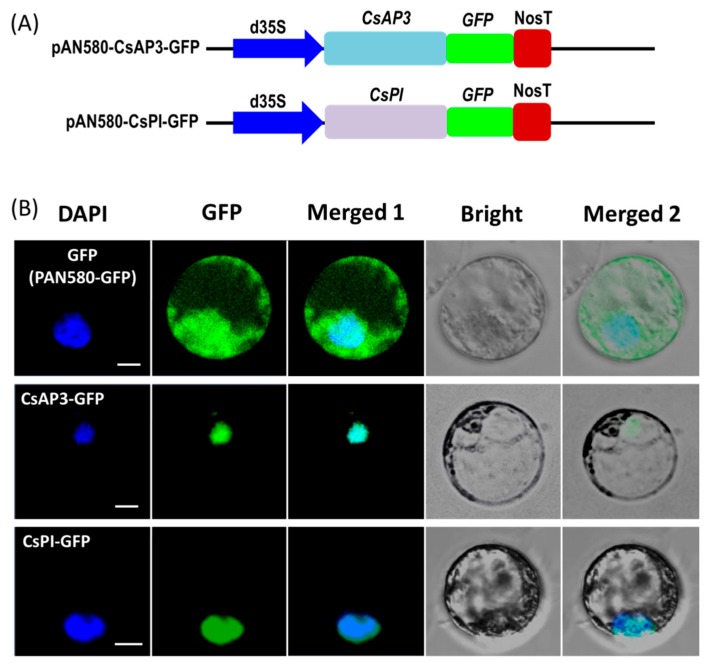
Protein subcellular localization studies using the PTES. (**A**) Vectors used for transient expression were obtained by cloning the full-length coding sequences (CDSs) of *CsAP3* and *CsPI* into the vector PAN580-GFP. (**B**) Recombinant vectors pAN580-*CsAP3*-*GFP* and pAN580-*CsPI*-*GFP*, as well as empty control vector (expressing GFP) were transformed into *Cymbidium* protoplasts to test feasibility of the PTES for protein subcellular localization; fusion proteins CsAP3-GFP and CsPI-GFP were both colocalized with the DAPI signal, indicating nuclear localization of both fusion proteins; bar = 10 μm.

**Figure 4 ijms-21-02264-f004:**
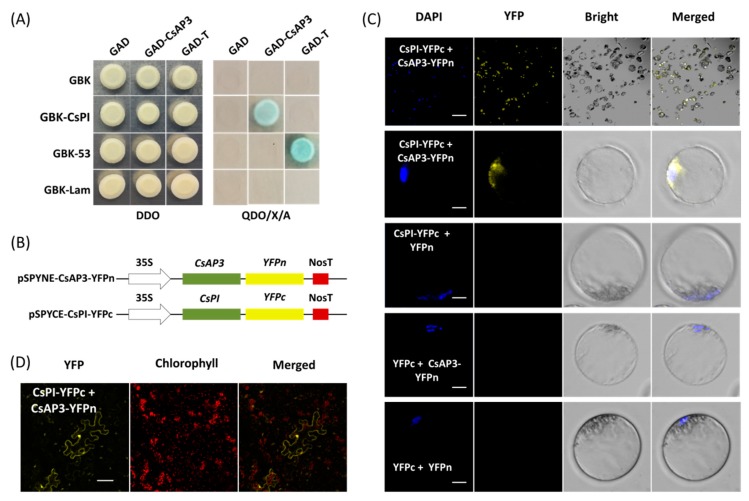
Protein–protein interaction studies using the PTES. (**A**) Yeast two-hybrid assay (Y2H) of proteins encoded by *CsPI* and *CsAP3*. (**B**) Vectors used for bimolecular fluorescence complementation (BiFC) assays were obtained by cloning the full-length CDSs of *CsAP3* and *CsPI* into the vectors pSPYNE-35S and pSPYCE-35S, respectively. (**C**) Fusion proteins CsAP3-YFPn and CsPI-YFPc were co-expressed in *Cymbidium* protoplasts and yellow fluorescent protein (YFP) signals were detected in nuclei where the DAPI signal presented, while negative controls did not produce any BiFC fluorescence; (**D**) BiFC assays were also carried out in N. benthamiana leaves and strong YFP signals were observed in the protoplasts, especially in the nuclei. Bar = 10 μm.

**Figure 5 ijms-21-02264-f005:**
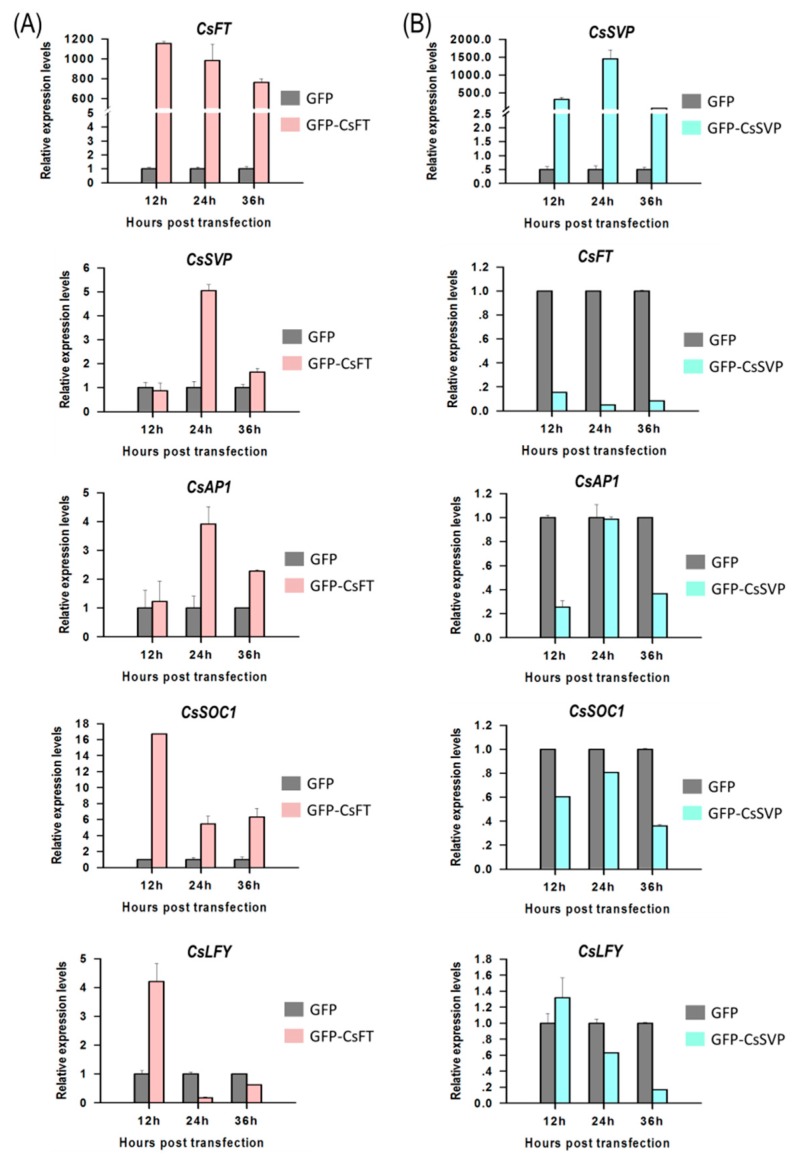
Gene regulation analysis using the PTES and qRT-PCR. (**A**) *CsFT* was successfully overexpressed in *Cymbidium* protoplasts using the PTES, which in turn resulted in upregulated expression of *CsSOC1*, *CsAP1*, *CsLFY* and *CsSVP* and (**B**) *CsSVP* was also successfully overexpressed in *Cymbidium* protoplasts 12–36 hpt, and the expression levels of *CsSOC1*, *CsAP1*, *CsLFY* and *CsFT* were significantly reduced. Y-axes indicate the relative expression levels of controls and transfected protoplasts at different time points (12, 24 and 36 hpt). Data were expressed as the mean of three biological replicates with error bars indicating the SD.

**Figure 6 ijms-21-02264-f006:**
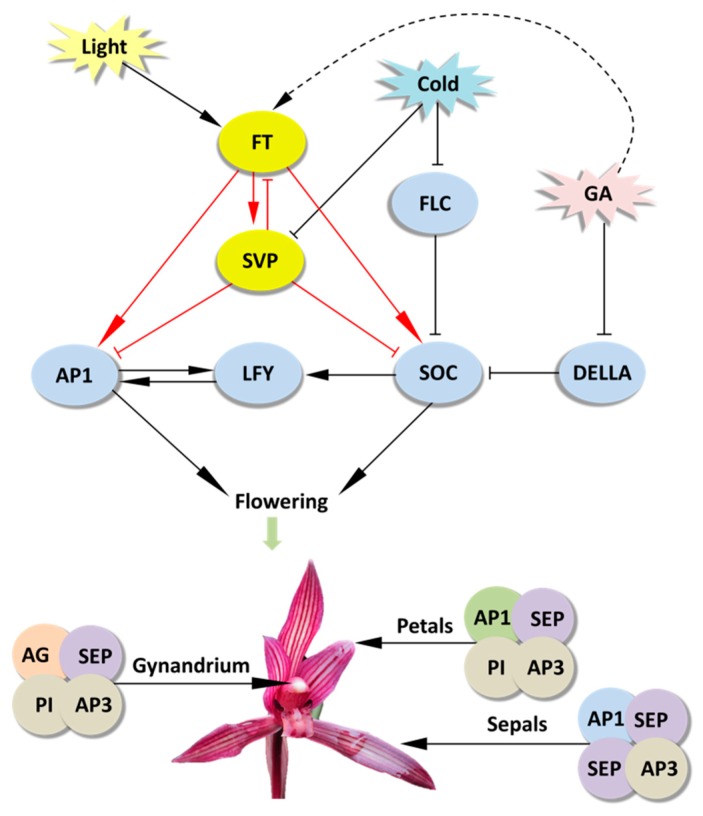
Conserved putative genetic regulatory networks of flowering in plants. Genes are represented by ovals. Lines with an arrow represent promotion, and those with a perpendicular bar represent repression. Red lines indicate regulation analyzed in this study.

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
