# Peer review of "Highly Efficient Protoplast Isolation and Transient Expression System for Functional Characterization of Flowering Related Genes in Cymbidium Orchids"

_ijms, 2020, doi:10.3390/ijms21072264_

Round 1

Reviewer 1 Report

This manuscript, entitled "Highly efficient protoplast isolation and transient expression system for functional characterization of flowering relevant genes in Cymbidium orchids" authored by Ren et al, presents an optimized protocol for high yield of protoplast isolation and transformation in Cymbidium orchids. They further demonstrated its use by transiently express flowering genes and testing for protein localization, protein-protein interaction by BiFC, induction of expression after ectopic expression of a TF and its analysis by qPCR.

The described work is interesting, well presented. However, I have a few comments.

  • Factors tested in Figure 2: three parameters are tested for optimization: PEG concentration, incubation time, DNA concentration. In the optimization, Figure 2B, what are the settings of the 2 fixed parameters when the third one is optimized? Or did you test all combinations possible? For example, optimizing plasmid concentration with 10% PEG or 25% PEG won't provide the same results. There is no mention of this in the text, figure legend or methods.
  • Format of figure 1: concerning the y-axis of protoplast viability in Fig 1B. It is %, why a scale of -10 to 110?
  • Figure 2C: the same protoplast picture is presented twice with transfection by two different plasmids. There is a problem.
  • Figure 4: All the interactions were tested in one way only. Did you test the reciprocal: GBK-CsAP3 / GAD-CsPI and CsPI-YFPn / CsAP3-YFPc?
  • Figure S2: Only some plasmid vectors are presented, not all. The figure title is misleading.
  • Spelling/typos:
    • Lines 182-184: CeAP3 > CsAP3 and CePI > CsPI
    • Line 247: form > from
    • Line 254: than that that > than the one

Author Response

#Reviewer 1

This manuscript, entitled "Highly efficient protoplast isolation and transient expression system for functional characterization of flowering relevant genes in Cymbidium orchids" authored by Ren et al, presents an optimized protocol for high yield of protoplast isolation and transformation in Cymbidium orchids. They further demonstrated its use by transiently express flowering genes and testing for protein localization, protein-protein interaction by BiFC, induction of expression after ectopic expression of a TF and its analysis by qPCR. The described work is interesting, well presented. However, I have a few comments.

Comment 1: Factors tested in Figure 2: three parameters are tested for optimization: PEG concentration, incubation time, DNA concentration. In the optimization, Figure 2B, what are the settings of the 2 fixed parameters when the third one is optimized? Or did you test all combinations possible? For example, optimizing plasmid concentration with 10% PEG or 25% PEG won't provide the same results. There is no mention of this in the text, figure legend or methods.

Answer: According to reviewer’s comment, we have revised the “Result 2.2” as “…The transfection efficiency increased from 29.19% to 79.09% with increasing PEG4000 concentration, and then decreased from 79.09% to 62.42% when the incubation time was 15 min, indicating optimal final PEG4000 concentration was 25%. Moreover, when given the optimal incubation time (15 min) and the final PEG4000 concentration (25%), the transfection efficiency increased significantly when using 1 - 2 μg/μL of plasmid DNA, and the efficiency remained at high level (~80%)…” (Result 2.2, Lines 150-152 in the revised manuscript).

Comment 2: Format of figure 1: concerning the y-axis of protoplast viability in Fig 1B. It is %, why a scale of -10 to 110?

Answer: The scale of y-axis of protoplast viability was automatically given as -10% to 110% in the software, and we are sorry for our carelessness. We have adjusted it (0-100%) according to the reviewer’s comment (Figure 1B).

Comment 3: Figure 2C: the same protoplast picture is presented twice with transfection by two different plasmids. There is a problem.

Answer: Many thanks for your detailed comments and corrections, and sorry for our carelessness. According to reviewer’s comment, we have changed the Figure 2C accordingly.

Comment 4: Figure 4: All the interactions were tested in one way only. Did you test the reciprocal: GBK-CsAP3 / GAD-CsPI and CsPI-YFPn / CsAP3-YFPc?

Answer: The interaction between proteins CsAP3 and CsPI were exactly tested in one way only in our yeast two-hybrid assay, because of the AP3 / PI interaction is well known in other plant species, such as rice (https://link.springer.com/article/10.1023/A:1025401611354) and Oncidium orchids (https://academic.oup.com/pcp/article/43/10/1198/1849356). In this study, we focused on the establishment and assessment the feasibility of the protoplast transient expression system. As it is important in regulation floral differentiation, the CsAP3 / CsPI interaction will be confirmed in another way such as GBK-CsAP3 / GAD-CsPI and CsPI-YFPn / CsAP3-YFPc, and using GST-pull down or Co-IP in our further study.

Comment 5: Figure S2: Only some plasmid vectors are presented, not all. The figure title is misleading.

Answer: The title of Figure S2 has been revised from “Plasmid vectors used in this study” into “Plasmid vectors used for gene regulation analysis in this study”. (Supplementary Materials, Line 477 in the revised manuscript).

Spelling/typos:

Comment 6: Lines 182-184: CeAP3 > CsAP3 and CePI > CsPI;

Answer: We have revised the word “CeAP3 and CePI” into “CsAP3 and CsPI”. (Results 2.4, Line 195 in the revised manuscript).

Comment 7: Line 247: form > from;

Answer: We have revised the word “form” as “from”. (Discussion, Line 259 in the revised manuscript).

Comment 8: Line 254: than that that > than the one

Answer: We have deleted the repeating “that”. (Discussion, Line 266 in the revised manuscript).

Reviewer 2 Report

In the paper titled "Highly efficient protoplast isolation and transient expression system for functional characterization of flowering relevant genes in Cymbidium orchids", Authors describe the development, the optimization and the assessement of a isolation/transfection new protocol of protoplast from petals of an orchid plant genus, Cymbidium.

This is a descriptive pretty jargon-dense paper, but even if in my research I have dealt with protoplasts just few times, I would say that the work seems sound. Several different molecular approaches, including BiFC and RT-qPCR on transformed protoplasts, were used to validate the results and this should be appreciated.

I have some comments/suggestions and few questions for the improvement of the paper, which can be considered for publication in IJMS after they will be addressed.

-First, it is unclear how BiFC assay was performed in N. benthamiana. In lines 193-194 it is claimed that YFP signal was observed in protoplasts, but in Methods it is stated that N. benthamiana leaves were used. Did Authors extract protoplasts from leaves? It is not so common in a BiFC with benthamiana. Please explain.

-In Figure 2C, the analyzed cells transformed with the two different GFP vectors (pAN580 and pCambia1301) look exactly the same. How was it possible? Did the showed cells were transformed with both plasmids? If yes, please include this information on methods and explain better how was done and why did not use single-plasmid transformation.

-Duncan test (lines 455-456) is a good one, but I would suggest to re-analyze the data by using Tukey test, which is most powerful in reducing bias due to type I errors.

Other minor comments:

-line 23: replace "via optimizing the factors" with "through the optimization of factors"

-line 27 and throughout the manuscript: BiFC is defined as Bimolecular fluorescence complementation and not "bimolecular fluorescent complimentary assay". This should be replaced by "Bimolecular fluorescence complementation (BiFC) assay".

-line 27: "related" better than "relevant"

-line 36: 1,500 to be consistent

-line 55: virgatum

-lines 56-59: split this sentence ("For plants.....been used") in two parts to be clear. As it is, it is hardo to follow.

-line 71: remove Crantz, or add authorship to all mentioned species to be coherent

-lines 78-79: "into protoplasts MEDIATED BY polyethylene glycol..."

-lines 91-93: better "and/or" as in other lines. Check throughout the manuscript.

-line 93: it is better to start with "It does not require..." and avoiding repetition in line 95 ("required").

-line 133, catpion: "statistically"

-line 185: PTES or PETS?

-line 192: N. bethamiana in italic

-line 240: I guess "through"

-Figure 6: I suggest to replace "demonstrated" with "analyzed" or "studied". The paper does not focused on demonstrating the role of these genes.

-Figure 5: Where are error bars for CsFT and CsSOC1? Please explain.

-In addition, a statistics on RT-qPCR results (ANOVA, use of REST2009; or so on) will be appreciated (but not mandatory, the focus of the work was not gene expression).

Author Response

#Reviewer 2

In the paper titled "Highly efficient protoplast isolation and transient expression system for functional characterization of flowering relevant genes in Cymbidium orchids", Authors describe the development, the optimization and the assessment of an isolation/transfection new protocol of protoplast from petals of an orchid plant genus, Cymbidium.

This is a descriptive pretty jargon-dense paper, but even if in my research I have dealt with protoplasts just few times, I would say that the work seems sound. Several different molecular approaches, including BiFC and RT-qPCR on transformed protoplasts, were used to validate the results and this should be appreciated.

I have some comments/suggestions and few questions for the improvement of the paper, which can be considered for publication in IJMS after they will be addressed.

Comment 1: First, it is unclear how BiFC assay was performed in N. benthamiana. In lines 193-194 it is claimed that YFP signal was observed in protoplasts, but in Methods it is stated that N. benthamiana leaves were used. Did Authors extract protoplasts from leaves? It is not so common in a BiFC with benthamiana. Please explain.

Answer: First, many thanks for your detailed comments, and sorry for our carelessness. In the updated version, the sentence “…strong YFP signals were observed in protoplasts…” has been corrected as “…strong YFP signals were observed in N. benthamiana mesophyll cells…”. (Result 2.4, Line 205 in the revised manuscript). Moreover, BiFC assay were generally carried out using a transient expression method. In this study, we firstly conducted the BiFC assay by co-transfection of CsAP3-YFPn, CsPI-YFPc and empty vector combinations in orchid protoplasts. The results indicated the interaction between proteins CsAP3 and CsPI. To confirm the interaction result, another transient expression method in N. benthamiana leaves was employed for BiFC assay, which has been used by Schütze et al. (2009, https://link.springer.com/protocol/10.1007/978-1-59745-289-2_12), Luan et al. (2019, https://www.mdpi.com/1999-4915/11/6/546) and Zong et al. (2020, https://www.sciencedirect.com/science/article/pii/S0168170219308081). As described in “Method 2.6 Line 424-429”, vectors CsAP3-YFPn and CsPI-YFPc were transformed into Agrobacterium tumefaciens strain GV301, respectively. And then positive agrobacteria which fused with reciprocal halves of YFP were co-infiltrated into N. benthamiana leaves. After dark-culture for two days, strong BiFC fluorescence was observed which indicating the expression of YFP and the interaction between CsAP3 and CsPI proteins.

Comment 2: In Figure 2C, the analyzed cells transformed with the two different GFP vectors (pAN580 and pCambia1301) look exactly the same. How was it possible? Did the showed cells were transformed with both plasmids? If yes, please include this information on methods and explain better how was done and why did not use single-plasmid transformation.

Answer: Many thanks for your detailed comments and corrections, and sorry for our carelessness. We have changed the figures. As described in Figure 2C caption, two different GFP vectors (pAN580 and pCambia1301) were respectively transfected into protoplasts, rather than co-transfection (two different vectors together). The transfections were carried out for so many times, and we noted that protoplasts would rupture or collapse with the increase of transfection efficiency. The biggest difference was that the maximum transfection efficiency of PAN-580 reached to ~80%, while that of pCambia1301-GFP was ~30%. The protoplasts transfected with distinct vectors look exactly the same as the “Reviewer 2” mentioned, was due to our mistakes in assembly of photos. This has been rectified. As shown in Figure 2C, the ratios of protoplasts transfected with PAN-580 displaying green fluorescence is indeed greater than that of pCambia1301.

Comment 3: Duncan test (lines 455-456) is a good one, but I would suggest to re-analyze the data by using Tukey test, which is most powerful in reducing bias due to type I errors.

Answer: We have re-analyzed the data by using Tukey–Kramer test (Figure1 B).

Other minor comments:

Comment 4: line 23: replace "via optimizing the factors" with "through the optimization of factors"

Answer: “via optimizing the factors” has been changed into “through the optimization of factors”. (Abstract, Line 26 in the revised manuscript)

Comment 5: line 27 and throughout the manuscript: BiFC is defined as Bimolecular fluorescence complementation and not "bimolecular fluorescent complimentary assay". This should be replaced by "Bimolecular fluorescence complementation (BiFC) assay".

Answer: We have checked throughout the manuscript according to reviewer’s comment, and changed all “bimolecular fluorescent complimentary” and “Bimolecular fluorescence complementation” to “Bimolecular fluorescence complementation”. (Abstract, Line 26; Keywords, Line 36; Result 2.4, Line 197; Abbreviations, Line 494; in the revised manuscript).

Comment 6: line 27: "related" better than "relevant"

Answer: “relevant” has been changed into “related” throughout the manuscript. (Title, Line 4; Abstract, Line 31; Result 2.3, Line 174; Discussion, Line 285, Line 288; in the revised manuscript).

Comment 7: line 36: 1,500 to be consistent

Answer: We have revised it to “2,000” which is more accurate rather than “1,500” (http://argusorchids.net/). (Introduction, Line 41 in the revised manuscript).

Comment 8: line 55: virgatum

Answer:Virgatum” has been changed to “virgatum”. (Introduction, Line 60 in the revised manuscript).

Comment 9: lines 56-59: split this sentence ("For plants.....been used") in two parts to be clear. As it is, it is hard to follow.

Answer: The sentence "For plants.....been used" has been split into two parts as “However, it is difficult to isolate protoplasts with high yield and viability from mature leaf tissues of many plants. Hence, young leaves of in vitro grown plantlets of Phalaenopsis [26], Dendrobium [27], Tanacetum [28], grape (Vitis vinifera) [29], pepper (Capsicum annuum L.) [30] and pineapple [31] have been used.”. (Introduction, Line 61-64 in the revised manuscript).

Comment 10: line 71: remove, or add authorship to all mentioned species to be coherent

Answer: Crantz (the Latin name of cassava) has been removed to keep the coherence of the sentence. (Introduction, Line 79 in the revised manuscript).

Comment 11: lines 78-79: "into protoplasts MEDIATED BY polyethylene glycol..."

Answer: The sentence "into protoplasts MEDIATED BY polyethylene glycol..." has been revised accordingly. (Introduction, Line 87 in the revised manuscript).

Comment 12: lines 91-93: better "and/or" as in other lines. Check throughout the manuscript.

Answer: "and/or" has been revised throughout the manuscript. (Introduction, Lines 87 and 96; Result 2.1, Line 111; in the revised manuscript).

Comment 13: line 93: it is better to start with "It does not require..." and avoiding repetition in line 95 ("required").

Answer: The sentence has been revised as “It does not require vacuum infiltration, treatment with high osmotic solution to plasmolyse tissues prior to enzymatic digestion, or special equipment.” (Introduction, Lines 101 and 103 in the revised manuscript).

Comment 14: line 133, caption: "statistically"

Answer: According to reviewer’s comment, “statically” has been changed into “statistically” in the manuscript. (Figure 1. caption, Line 142 in the revised manuscript).

Comment 15: line 185: PTES or PETS?

Answer: “PETS” has been revised into “PTES”. (Results 2.4, Line 196 in the revised manuscript).

Comment 16: line 192: N. bethamiana in italic

Answer: According to reviewer’s comment, “N. bethamiana” has been changed to “N. bethamiana”. (Results 2.4, Line 205 in the revised manuscript).

Comment 17: line 240: I guess "through"

Answer: “though” has been revised into “through”. (Discussion, Line 252 in the revised manuscript).

Comment 18: Figure 6: I suggest replacing "demonstrated" with "analyzed" or "studied". The paper does not focused on demonstrating the role of these genes.

Answer: “demonstrated” has been revised into “analyzed”. (Figure 6, Line 301 in the revised manuscript).

Comment 19: Figure 5: Where are error bars for CsFT and CsSOC1? Please explain.

Answer: Error values for CsFT and CsSOC1 are relatively small, because of the data of three biological replicates were almost similar. Error bars could be seen, if the picture locally is enlarged (Figure 5).

Comment 20: In addition, a statistics on RT-qPCR results (ANOVA, use of REST2009; or so on) will be appreciated (but not mandatory, the focus of the work was not gene expression).

Answer: Actually, we have done variance analysis and assigned many schematic symbols in the original manuscript. But some authors felt it looked too complicated and we decided to simplify the figures and removed the symbols for the differences which are obvious for some genes at certain time points (Figure 5).